# WeNet-RF: An Automatic Classification Model for Financial Reimbursement Budget Items

**Peichun Suo[1], Xiuyan Wang[2]\*, Weili Kou[3]\*, Wen Suo[1], Yujing Zhang[4], Jinfen Duan[5], Tingting Zeng[1], Meicai Zhu[1], Fubing Wang[1]**

**1** Yunnan Medical Health Vocational College, Kunming, Yunnan, China, **2** Yunnan College of Business Management, Kunming, Yunnan, China, **3** College of Big Data and Intelligent Engineering, Southwest Forestry University, Kunming, Yunnan, China, **4** Sangfor Technologies Inc, Kunming, Yunnan, China, **5** Yunnan Polytechnic College, Kunming, Yunnan, China

\* 1641197926@qq.com (X.W.); kwl@swfu.edu.cn (W.K.)

## Abstract

Accurate classification of budget items is a critical component of financial reimbursement, as it determines the legitimacy and regulatory compliance of financial expenditures. Currently, manual classification of reimbursement budget items faces to two challenges of inefficiency and inaccuracy. This is primarily due to the labor-intensive nature of the task, which increases the likelihood of selecting incorrect categories. To address these challenges, this study proposed a WeNet-Random Forest (WeNet-RF) model, which leverages speech recognition technology (WeNet) and Random Forest (RF) to improve efficiency and classification accuracy. WeNet-RF includes four steps: speech identification, features extraction, items classification, and evaluated model. This study compared WeNet-RF with Convolutional Neural Networks (CNN), Logistic Regression (LR) and K-Nearest Neighbors (KNN). WeNet-RF was verified by 50 real financial reimbursement records, and the results show that accuracy rate, precision rate, recall rate, and F1 score of WeNet-RF all are 90.77%. The findings provide a robust solution for improving financial management processes, and a reference model to financial management system.

## 1. Introduction

Automated categorization of financial reimbursement expenses is essential for improving operational efficiency and accuracy in corporate finance management. Automatic classification of financial reimbursement budget items is crucial for enhancing the efficiency and accuracy of corporate financial management [1]. By reducing manual operations, it minimizes classification errors and time costs while ensuring consistency and compliance. Automated classification improves the employee reimbursement experience and enables finance teams to allocate more resources to strategic tasks such as resource optimization and performance evaluation. Current financial reimbursement processes involve manual data entry, relying

**Data availability statement:** Due to the requirements of data confidentiality, the dataset of the school's financial department can be obtained by contacting the school office via email. (Name: Tingting Zeng; Contact number: +86 871-68305391; Email address: 1908584074@qq.com).

**Funding:** This research was supported in part by research grants from the "Yunnan Revitalization Talent Support Program" in Yunnan Province (grant NO. YNWR - QNBJ - 2019 - 270), and the Scientific Research Fund of Yunnan Provincial Department of Education under Research and Development Project (grant NO. 2024J214).

**Competing interests:** The authors have declared that no competing interests exist.

heavily on handwritten forms, which not only requires significant time and effort from claimants but also leads to errors in matching expenses with budget categories. This frequently results in repeated submissions for corrections, substantially impacting work efficiency. The primary issues currently affecting the efficiency and accuracy of financial budget item classification in financial reimbursement are the entry of reimbursement information and the identification of item classification. With the advancement of artificial intelligence technology, developing an automated classification model for financial reimbursement items is one of the critical methods to address this problem.

The application of machine learning in finance has increasingly become a significant focus in both academic research and business practice. Notably, the development of financial models utilizing machine learning is propelling financial management into a new era. These models capitalize on machine learning techniques by leveraging big data and advanced computational capabilities to predict the digital behaviors of bank customers based on comprehensive consumer finance surveys [2]. This innovation has fundamentally transformed various areas, including financial efficiency scoring, customer flow analysis, risk assessment, and investment decision-making [3,4]. In addition, autocompletion technology has simplified manual form-filling processes, enhancing both efficiency and accuracy while yielding substantial benefits across multiple industries. Its extensive application value and profound impact are evident in key sectors such as census data collection, meteorological observation, virtual reality environments, pharmaceutical production, psychological assessment, and high-aviation operations. The adoption of autocompletion has facilitated rapid and precise data entry, significantly improving work efficiency and data quality. Comparative studies on error rates between autocompletion models and manual filling methods in wheat stacking experiments have highlighted the advantages of automation [5]. In the context of contour cutting, automated algorithms have successfully executed solid sectioning, automatic filling, and material inheritance of 3D models within virtual reality environments [6]. Moreover, innovative methods for addressing data gaps in monthly MODIS (Moderate Resolution Imaging Spectroradiometer) land surface temperature datasets, often obscured by clouds and atmospheric disturbances, have been proposed [7]. Another method focuses on the precise positioning of filling and sealing robots using Gaussian Mixture Models, particularly advantageous in pharmaceutical production [8]. Research has also examined the effects of automated filling equipment on the dispersion of various pharmaceutical formulations [9]. Furthermore, in psychological assessments and high-aviation operations, a model for the automatic filling of the Beck Depression Inventory questionnaire has been developed, underscoring the potential of automation in psychological evaluations [10]. Optimization of the filling process through machine settings adjustments demonstrates how mathematical modeling and linear programming can effectively minimize filling times for different orders [11].

Recent years have witnessed remarkable advancements in speech recognition applications, particularly in terms of accuracy and efficiency. The introduction of the Multiple Input Multiple Output Speech (MIMO-Speech) architecture has enabled

multi-speaker speech recognition [12]. Furthermore, the significance of large language models for achieving high accuracy has been emphasized [13]. The critical role of feature extraction algorithms in enhancing speech emotion recognition rates has also been highlighted [14], alongside efforts to optimize feature parameter extraction methods [15]. Additional research has explored unsupervised cross-corpus speech emotion recognition [16], proposed models for age and gender recognition [17], and introduced the WeNet toolkit as an efficient Automatic Speech Recognition (ASR) solution [18]. Further studies focus on the speech recognition capabilities of cochlear implant recipients [19] and the development of end-to-end speech benchmarks [20]. Simultaneously, the FunASR open-source toolkit has been launched to bridge the divide between academic research and industrial applications [21]. Collectively, these initiatives advance speech recognition across multiple domains and expand its practical applications.

In this study, we employ the Random Forest algorithm, a robust ensemble learning model. The prevalence of automatic classification algorithms in the financial sector is increasing, as they provide significant advantages in terms of enhanced efficiency and accuracy. In the context of personal loans and enterprise investments, Random Forests can accurately assess the creditworthiness of individuals or organizations by conducting a comprehensive analysis of multi-dimensional data, thereby offering strong support for loan approvals and investment decisions [22]. In financial analysis, Random Forests facilitate the automated classification and interpretation of financial statements, assisting investors and business managers in quickly grasping the financial health and performance of enterprises [23]. Moreover, Random Forests demonstrate considerable potential for financial forecasting, illustrated by the classification and prediction of customer financial data [24], making them a powerful tool for market analysis and strategic development.

Constrisution of this study includes: (1) Addressing the limitations of manual classification, this study proposed a WeNet-RF model to tackle the inefficiency and inaccuracy associated with manual classification of financial reimbursement budget items. By integrating speech recognition technology with the RF algorithm, the model significantly enhances classification efficiency and accuracy. (2) This study provides an efficient and reliable solution and framework for the automated classification of financial reimbursement budget items. The paper is organized as follows: Section 1 reviews the related work. Section 2 details the methodology. Section 3 presents the experimental results. Section 4 discusses the findings. Section 5 highlights the limitations of the model. Section 6 concludes the study. Section 7 outlines future work.

## 2. Research methods

### 2.1. Data and preprocessing

**2.1.1. Data sources description.** This study was approved by the research institution of Yunnan private vocational college. All participants provided informed consent before participating in this study and were fully informed about the purpose, potential risks, and benefits of the research. Participants indicated that they fully understood and voluntarily agreed to participate in the study.We obtained written consent. All participants signed written consent forms, and these documents have been securely stored to ensure the integrity of the data and the protection of participants' privacy.

The data for this study were obtained from the database of the financial integration management platform at a private vocational college in Yunnan Province, covering the period from January 1, 2022, to December 31, 2022. We collected and collated various data sources, including five voice files captured using three widely used models of mobile devices within the school. Additionally, we captured data from over 100 budget items across the university, encompassing 15 different approval processes and a total of 7,959 records of reimbursement information. The study also collected basic information on more than 650 faculty members. Regarding budget items, the budget item table recorded detailed information for each item, containing a total of 17 fields, as presented in Table 1. Among these fields, eight are of varchar type for storing textual information, while nine are of float type for storing numerical data.

The comprehensive budget item table comprises 806 records, including 100 budget items and 706 extrabudgetary items.These items encompass 14 secondary colleges and functional departments within the university, highlighting the intricacy and heterogeneity of the institution's financial management. The structure of the reimbursement account

**Table 1. Budget Items.**

| Field Name | Type | Instructions | Field Name | Type | Instructions |
|---|---|---|---|---|---|
| Item Code | String | Code Unique | Borrowings at end of year | Float | Borrowings at end of year |
| Item Name | String | Item Name | Number of freezes | Float | Number of freezes |
| Beginning balance | Float | Beginning balance | available balance | Float | available balance |
| Borrowings at beginning of year | Float | Borrowings at beginning of year | Item Category Codes | String | Item Category Codes |
| Income incurred in the year | Float | Income incurred in the year | Item Category Name | String | Item Category Name |
| Expenditure incurred during the year | Float | Expenditure incurred during the year | Departments | String | Departments |
| Borrowings incurred during the year | Float | Borrowings incurred during the year | responsible for manual numbering | String | responsible for manual numbering |
| Adjustments incurred during the year | Float | Adjustments incurred during the year | Name of person in charge | String | Name of person in charge |
| Year-end balance | Float | Year-end balance | | | |

information table utilized in the study consists of a total of 80 fields, as illustrated in Table 2. Among these fields, 33 are of the VarChar data type, employed for storing textual information; 21 fields are of the float data type, used for storing numerical data; 1 field is of the Boolean data type, utilized for recording binary status information such as yes/no or true/false; and 3 fields are of the Date data type, used for recording date-related information.

The table comprehensively presents 7,959 reimbursement records, illustrating the intricacy and variety of the school's financial operations. The study has been approved by the institution's finance department, validating the data's reliability and facilitating the research's progression.

During the speech recognition session, 30 representative speech samples were collected to cover diverse reimbursement needs across various departments and projects, reflecting an array of real-world reimbursement scenarios. The file format was standardized as ".wav" with a maximum size of 5,120KB. All recordings were in Mandarin Chinese, focusing on key details such as the individual involved, time, location, nature of the completed task or activity, incurred expenses, and the reimbursement amount processed. To ensure clarity, each recording was limited to a duration of less than one minute.

**2.1.2. Research data.** The data employed in this study originates from the financial integrated management platform database of a private vocational college in Yunnan, covering the timeframe from January 1, 2022, to December 31, 2022. The complete dataset consists of 13,539 reimbursement records. Following a thorough filtering process, 7,959 data points were selected for model training and validation, representing various categories, including budget items and reimbursement details. To facilitate the development of the model, the dataset was partitioned into training and test sets in an 80:20 ratio. Approximately 6,367 data points were designated for training the Random Forest-based classification model for reimbursement budget items, while 1,592 reimbursement entries were allocated for testing purposes.

**2.1.3. Data preprocessing.** For speech processing, we converted the file formats, adjusted sample sizes, and standardized naming conventions for the speech sample files used in the study. To ensure the accuracy and reliability of subsequent analyses, we carefully addressed missing values in the financial claims data. We deleted 706 non-regular budget items coded outside the financial budget to prevent interference with the analysis results. Simultaneously, we imputed missing values in other budget items using the most probable values, particularly for key information such as budget items name, responsible person, and summary. For reimbursement dataset processing, we first preprocessed the original 80 columns of data, retaining 30 columns after deleting null-value columns. These 30 columns include 22 character type columns (10 character type and 12 classification features), 6 numerical value columns (day, sequence, debit amount, credit amount, debit quantity, and credit quantity), and 2 date-time type columns (billing time and review time). Second, we focused on missing data in the fields of reimbursement item code,

**Table 2. Reimbursement Account Information Sheet.**

| Field Name | Type | Instructions | Field Name | Type | Instructions |
|---|---|---|---|---|---|
| Number | String | number | cashier | String | cashier |
| Day | Int | day | Bank teller | String | Bank teller |
| Serial number | Int | serial number | Write-off code | String | Write-off code |
| Reservation number | String | reservation number | Correspondents | String | Correspondents |
| Typology | String | typology | creditor and debtor | String | creditor and debtor |
| Accountants | String | accountants | settlement terms | String | settlement terms |
| Item Code | String | Item Code | cheque number | String | cheque number |
| Item Name | String | Item Name | Date of operation | Date-Time | Date of operation |
| Basic subject | String | basic subject | other side's account name | String | other side's account name |
| Economic subject | String | economic subject | Counterparty Account Number | String | Counterparty Account Number |
| Income and expenditure type code | String | Income and expenditure type code | use | String | use |
| Name of income and expenditure type | String | Name of income and expenditure type | account opening bank | String | account opening bank |
| Abstracts | String | abstracts | UNB | String | UNB |
| Agent | String | agent | bank account number | String | bank account number |
| Debit amount | Float | Debit amount | source code | String | source code |
| Credit amount | Float | credit amount | Source name | String | Source name |
| Whether or not to flush red | Bool | Whether or not to flush red | appropriation number | String | appropriation number |
| Subproject Code | String | Subproject Code | Financial projects | String | Financial projects |
| Sub-project name | String | Sub-project name | Financial and economic subjects | String | Financial and economic subjects |
| Main Project Code | String | Main Project Code | payment order | String | payment order |
| Name of the main project | String | Name of the main project | Expenditure type code | String | Expenditure type code |
| Authorization number | String | authorization number | Nature of funding | String | Nature of funding |
| Budget items code | String | Budget items code | Account Type | String | Account Type |
| Budget items name | String | Budget items name | Category | String | Category |
| Billing | String | billing | budget year | String | budget year |
| Time of making order | DateTime | time of making order | Information association number | String | Information association number |
| Reconsider | String | reconsider | special accounting code | String | special accounting code |
| Review period | DateTime | review period | Special accounting designations | String | Special accounting designations |
| Material number | String | Material number | vehicle registration number | String | vehicle registration number |
| Number of debits | Float | Number of debits | note | String | note |
| Number of credits | Float | Number of credits | Broad categories of projects | String | Broad categories of projects |
| Current budget items | String | Current budget items | responsible for manual numbering | String | responsible for manual numbering |
| Contract number | String | contract number | Name of person in charge | String | Name of person in charge |
| SA_DEPART | String | Name of person in charge | Freeze Code | String | Freeze Code |
| SA_F01 | String | standby | SA_F02 | String | standby |
| SA_F03 | String | standby | SA_F04 | String | standby |
| SA_F05 | String | standby | SA_F06 | String | standby |
| SA_F07 | String | standby | SA_F08 | String | standby |
| SA_F09 | String | standby | SA_F10 | String | standby |
| SA_F11 | String | standby | SA_F12 | String | standby |

main item name, summary, reimbursement amount, item manager, reimbursement name, and reimbursement manual number. We deleted some of the 30 columns containing null values and vectorized the 12 key-value queue categorical features using Term Frequency-Inverse Document Frequency (TF-IDF), removing 6 key categorical features (booking/ budget type, 37 budget item names, whether or not to redress, 40 account names, 140 main item names, and 24 broad item categories). As the impact of missing values in attachment fields is relatively small, we deleted rows or columns containing missing values. For missing reimbursement claimant name fields, we populated them using the reimbursement claimant's job number field to ensure data integrity and consistency. After constructing the text dataset containing budget items and summaries, we preprocessed the dataset by removing irrelevant information from the summaries, de-duplicating data, and standardizing formatting. Ultimately, we retained 14 columns as characteristic columns, including summary, account name, operator, whether to flush red, sub-project name, main project name, budget item name, billing, billing time, review, review time, major category of the project, name of the person in charge, and department. The main steps in processing the financial claims dataset include text cleaning, word splitting, and removal of deactivated words, aiming to improve data quality and consistency and provide strong support for subsequent analysis and modeling. Text cleaning effectively removes irrelevant information and noise from the text, making the data more pure and useful [25]. Employing the jieba tool for word segmentation and stopword removal helps extract key features in the text, providing more valuable information for subsequent classification tasks. These optimization measures enhance the classification performance and adaptability of the model to real-world business. When dealing with texts involving complex financial terms or professional knowledge, preprocessing can significantly improve the performance of the classification model, such as for summary content like "income and expenditure," "financial income," and "special expenditure".This finding demonstrates the important role of preprocessing in improving text data quality and classification model performance. In this study, we used WeNet to convert speech into text. We used the accurate mode of jieba to segment the text and delete stop words. We also used the TF-IDF method to vectorize the text and convert it into a term frequency matrix.Finally,we collected 13,539 financial reimbursement data points. After rigorous screening, we excluded 5,580 invalid data points and identified 7,959 data points for model training and validation, as shown in Fig 1. The retained 7,959 data points are more effective in ensuring data accuracy and reliability, providing a solid foundation for subsequent analyses.

## 2.2. Modeling ideas

This study aims to address the challenges of rapid voice input for reimbursement information and the accuracy and efficiency of budget item classification. To this end, we proposed a financial reimbursement budget item classification model that integrates the WeNet framework and RF, as illustrated in Fig 2. We evaluate the performance of the RF-based model by comparing it against CNN, LR, and KNN models using an identical dataset.

To validate the performance of the RF model, we evaluated 50 real financial reimbursement information entries for voice entry and budget items classification. We utilized evaluation metrics such as accuracy, precision, recall, and F1 score to comprehensively measure the model's performance. The experimental results demonstrate that the RF model performs well across all evaluation metrics, with accuracy, precision, recall, and F1 score reaching 90%, effectively improving the accuracy and efficiency of reimbursement information entry and budget item classification. This study not only provides an effective model for the rapid entry of financial reimbursement information and accurate classification of budget items but also offers novel insights and methodologies for the application of machine learning in the financial domain.

## 2.3. Modeling step

The construction of the model in this study consists of four steps, as illustrated in Fig 3. The first step involves project reimbursement voice information acquisition and preprocessing. Speech signals are captured through microphones and

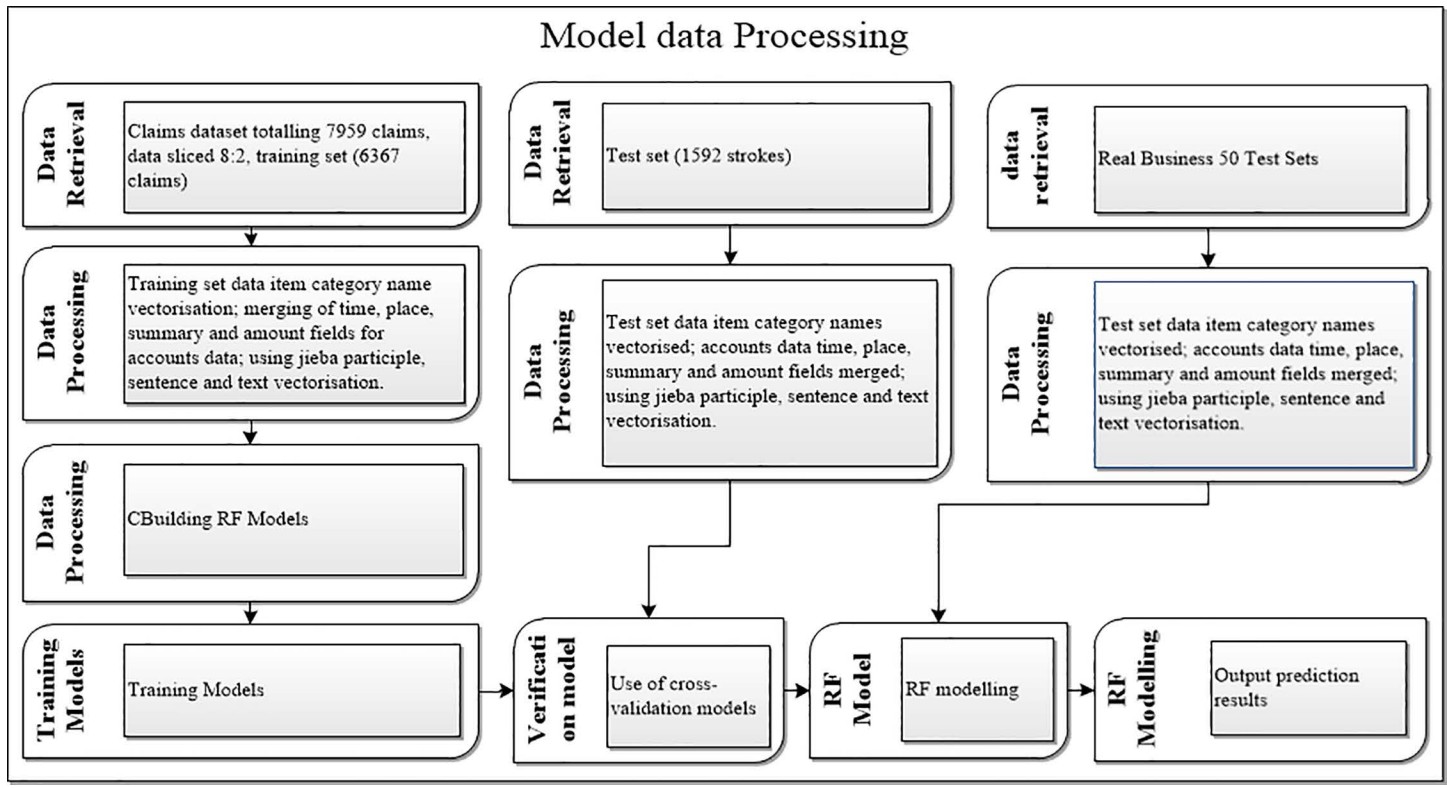

**Fig 1. RF Model Data Processing.**

other devices, followed by preprocessing operations such as noise reduction and removal of repetitions to enhance the quality of the speech signals. Subsequently, the WeNet open-source framework is employed to automatically convert speech to text with speed and accuracy. In the second step, we preprocessed the project reimbursement dataset. The 2022 financial budget project and reimbursement dataset undergoes deduplication, and Chinese word splitting is accomplished using jieba. Text vectorization is performed, converting the text to a word frequency matrix based on TF-IDF [26]. The third step involves constructing a financial reimbursement budget item classification model based on RF. In the fourth and final step, the study is evaluated using the test dataset.

## 2.4. Feature selection

The model indicators include two core elements: summary information on financial claims and budget items. The accuracy and completeness of the summary, which provides a general description of the reason for the reimbursement, is essential for a thorough understanding of the reimbursement's content. Additionally, the correct classification of the budget items is crucial for ensuring reimbursement compliance and the accuracy of financial processing. In our comprehensive exploration of the principles of financial reimbursement standardization, we had clearly stipulated that each reimbursement operation must be thoroughly documented with the following information: the reason for the reimbursement, the expense item, the time the expense was incurred, the specific location, the total reimbursement amount, and details on the reimbursement and approving officer. These core elements collectively form the basic indicator system of reimbursement information, effectively guaranteeing the transparency and traceability of financial activities. Among these indicators, the summary, expense item, time, location, amount, and reimbursement person information are considered key indicators

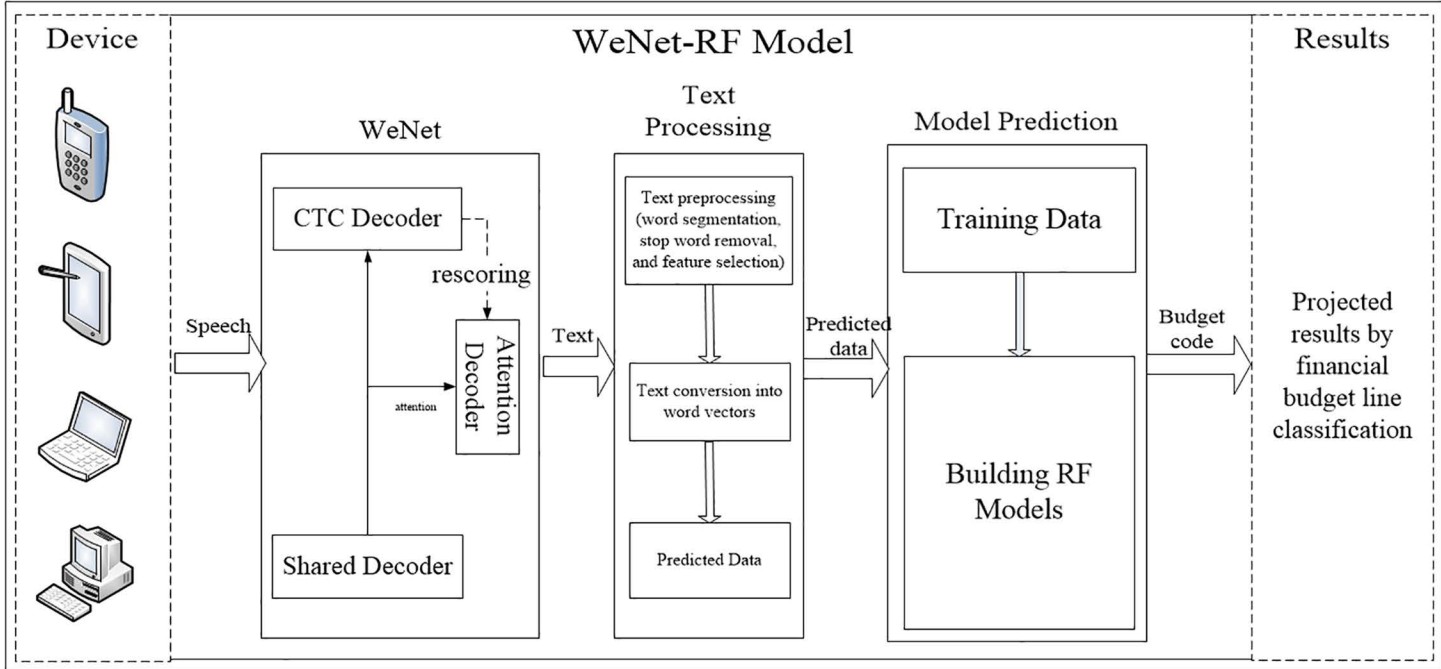

**Fig 2. A WeNet and RF-based Classification Model for Financial Reimbursement Budget Items.**

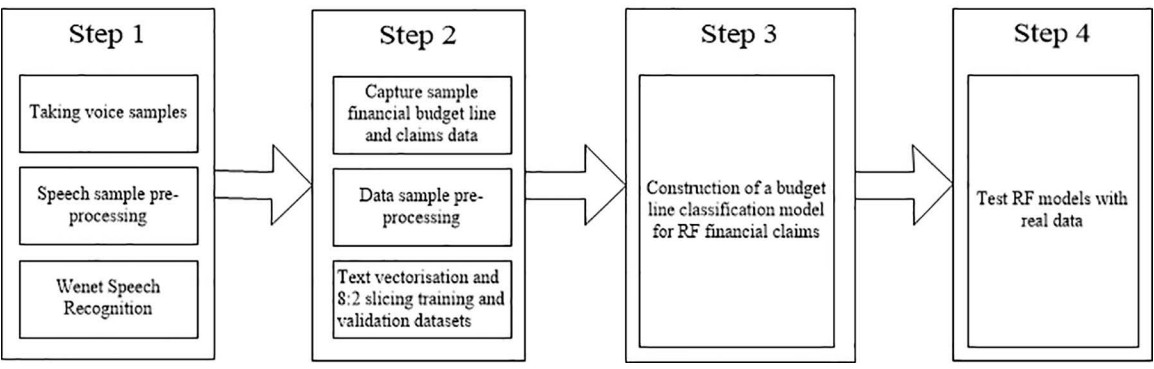

**Fig 3. WeNet-RF Model Modelling Idea.**

that require accurate input by the operators to precisely reflect the details of the reimbursement activities. The approver's information is subsequently processed through system automation, which enhances processing efficiency and minimizes human errors.

To streamline the dataset, we employed the Recursive Feature Elimination (RFE) technique during feature selection. This method iteratively constructs the model and removes the least important features. The preprocessing stage retained 14 feature columns, including summary, subject name, operator, whether to flush red, sub-project name, main project

name, budget item name, billing, billing time, review, review time, major project category, person in charge, and the associated department. Using Random Forest as the base model, RFE gradually eliminated minor attributes and ranked them according to feature importance, as illustrated in Fig 4. The process incorporated key items from the financial reimbursement information, such as the reimbursement project, summary, amount, operator (auto-filled by the system), time (auto-filled by the system), and the operator's department (pre-budgeted by the system). Ultimately, 5 features – summary, main project name, operator, project leader, and department – were selected for model construction.

## 2.5. WeNet

The WeNet speech recognition system offers considerable advantages for the automatic completion of financial reimbursement information. Its high recognition accuracy and stability ensure that verbal reimbursement details can be swiftly and accurately converted into written text, even in complex environments [27]. By employing an end-to-end training model with multi-layer transformers or conformers to capture deep features, WeNet achieves efficient recognition and text conversion. Employees can input reimbursement information through speech, and the system automatically populates the relevant fields in reimbursement forms, effectively minimizing handwriting errors and preventing incomplete submissions. This capability significantly enhances the efficiency of information entry. Furthermore, by integrating machine learning models, WeNet can automatically classify and review reimbursement information, streamlining the process and improving overall work efficiency. This integration fosters the modernization and intelligent management of corporate financial operations.

## 2.6. Random forest model

Random Forest (RF) is an ensemble learning model that integrates multiple individual decision trees, excelling in both classification and regression tasks. Its primary advantages include exceptional accuracy, a low propensity for overfitting, and robust performance across diverse scenarios. By constructing and aggregating numerous decision trees, RF optimizes model performance and effectively manages a large array of features as well as nonlinear relationships [28]. In this study, RF was optimized through grid search, k-fold cross-validation (k = 5, the data is shuffled and the random seed is set to 200) is used to train and evaluate the random forest classifier. Finally, the RandomForestClassifier was configured with the following parameters: n_estimators=1000 (indicating that the model consists of 1,000 decision trees), max_features=5 (which limits the model to consider up to 5 features for each tree), and random_state=200 (to ensure the reproducibility of

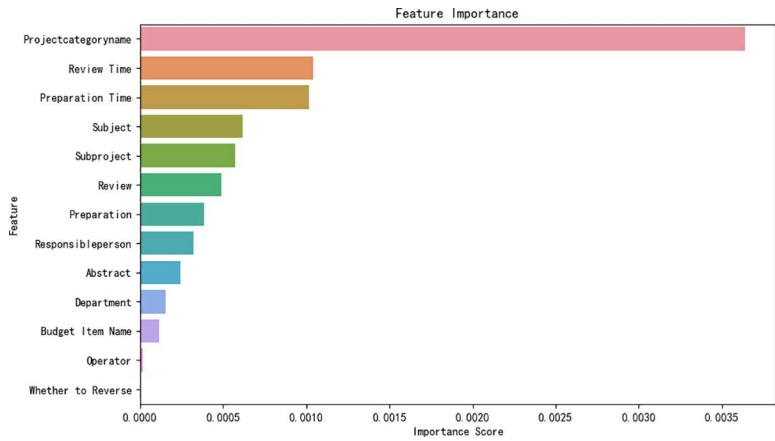

**Fig 4. Ranking of Feature Importance.**

results through a fixed random seed). Furthermore, this study conducted a comparative analysis of the RF method against Convolutional Neural Networks (CNN), Logistic Regression (LR), and K-Nearest Neighbors (KNN). The CNN model was developed using the PyTorch framework, featuring three Block instances with varying convolutional kernel sizes. It was equipped with an embedding matrix, a fully connected classification layer, and a cross-entropy loss function [29]. The LR model was implemented through the LogisticRegression class, utilizing the parameter random_state=200 to maintain consistency, and offering strong probabilistic interpretability along with high computational efficiency [30]. The KNN algorithm was executed using the KNeighborsClassifier, with n_neighbors set to 5, meaning the classifier would consider the five nearest neighbors when making classification decisions [31]. While KNN is effective for handling multi-class classification problems, it can become computationally expensive and sensitive to noise and irrelevant features, particularly when applied to large datasets [32]. The experimental results revealed that RF outperformed the other methods in terms of classification accuracy, robustness, and its capacity to manage high-dimensional features, particularly in addressing complex nonlinear challenges. Therefore, RF has been established as the optimal choice for a range of machine learning applications.

### 2.7. Cross-validation assessment

Cross-validation [33] is an effective model evaluation method for assessing the generalization ability and stability of a model. In the evaluation of RF models, we employed cross-validation to evaluate model performance. The model was trained 5 times, with a different random seed used each time. The results are then averaged. First, the dataset was divided into subsets, each containing a specific number of samples. Then, k-1 of these subsets were used for training, while the remaining subset was used for testing. This process was repeated k times, with each iteration selecting a different subset as the test set and the remaining subsets as the training set. In each iteration, we calculated the model's performance metrics, such as accuracy, recall, and F1 score. Through cross-validation, we obtained the model's performance on several different datasets, providing a more comprehensive understanding of its generalization ability. Furthermore, cross-validation can assist in identifying potential overfitting or underfitting issues in the model. In practice, we utilized k-fold cross-validation to evaluate the automatic entry of financial reimbursement information. The dataset was divided into 10 subsets, with k-1 subsets used for training and the remaining subset for testing in each iteration. By comparing the performance metrics across different subsets, we obtained the accuracy of the four models, as depicted in Fig 5.

Cross-validation evaluation revealed that the RF and LR models exhibited high accuracy and stability in the prediction task. The RF model effectively mitigates overfitting by integrating multiple decision trees while maintaining robust classification capabilities. Conversely, the LR model has demonstrated strong performance in binary classification tasks due to its simplicity and efficiency [34]. Both models displayed consistent prediction results during cross-validation, providing compelling evidence for the practical application of financial reimbursement expense item selection.

## 3. Analysis of results

### 3.1. Experiment results

Accuracy is a critical metric for assessing the performance of classification models, representing the proportion of correctly predicted samples [33,35]. Recall indicates the model's effectiveness in identifying key information [2,33], while the F1 score offers a comprehensive evaluation of the model's overall performance [33].

In this study, the RF model was compared with CNN, LR, and KNN. The accuracy, precision, recall, and F1 score of the RF-based reimbursement budget item classification model all exceeded 90%, demonstrating its superior performance in this context. Based on a validation set of 1,592 reimbursement data points and a real test set of 50 instances, Table 3 presents a comparison of the RF model's performance against CNN, LR, and KNN models. The RF model achieved an accuracy, precision, recall, and F1 score of 90.77% across all metrics. The CNN model exhibited the weakest performance, with accuracy, precision, recall, and F1 score of 22.27%, representing a difference of 68.5% compared to the

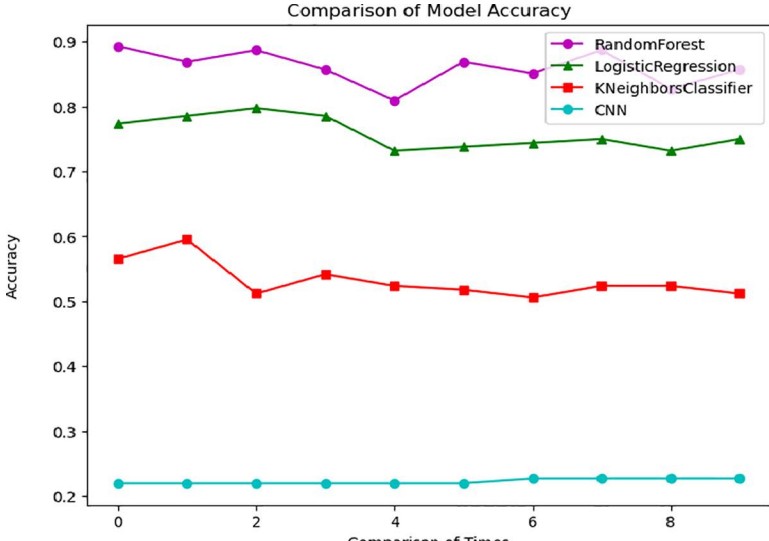

**Fig 5. Training Set Cross-Validation Evaluation.**

**Table 3. Experimental Results in the Validation Set.**

| Model | Accuracy | Precision | Recall | F1 score |
|---|---|---|---|---|
| CNN | 0.2227 | 0.2227 | 0.2227 | 0.2227 |
| LR | 0.8601 | 0.8601 | 0.8601 | 0.8601 |
| RF | 0.9077 | 0.9077 | 0.9077 | 0.9077 |
| KNN | 0.7369 | 0.7369 | 0.7369 | 0.7369 |

RF model. The LR model demonstrated the second-best performance, with accuracy, precision, recall, and F1 score of 86.01%, indicating a difference of 4.76% relative to the RF model. The KNN model ranked third, achieving an accuracy, precision, recall, and F1 score of 73.69%, with a difference of 17.08% compared to the RF model and 51.42% compared to the CNN model, as illustrated in Fig 6. To assess the generalization ability of the RF model, it was evaluated using 50 real business data instances, yielding an accuracy, precision, recall, and F1 score of 90.77%.

In this study, all indicators of the RF model surpass 90%, demonstrating its ability to accurately identify and extract key information from financial reimbursement data. This high performance suggests that the RF model possesses strong classification capabilities and can comprehensively identify essential details within the summary text. When compared to CNN, LR, and K-Nearest Neighbors (KNN) models, the RF model exhibits significant advantages in terms of accuracy, precision, recall, and F1 score [36], as illustrated in Fig 6. These findings indicate that the stochastic deep forest model can effectively learn and understand complex patterns in Chinese text, even when applied to small datasets. The RF model automatically extracts features from the text in the reimbursement summary information and utilizes these features for classification and prediction.

## 3.2. Case analysis

For example, in the case of a medical higher vocational college in Yunnan with a departmental budget of less than 10,000 yuan, the school currently uses an app or webpage for financial reimbursement information entry. Each reimbursement

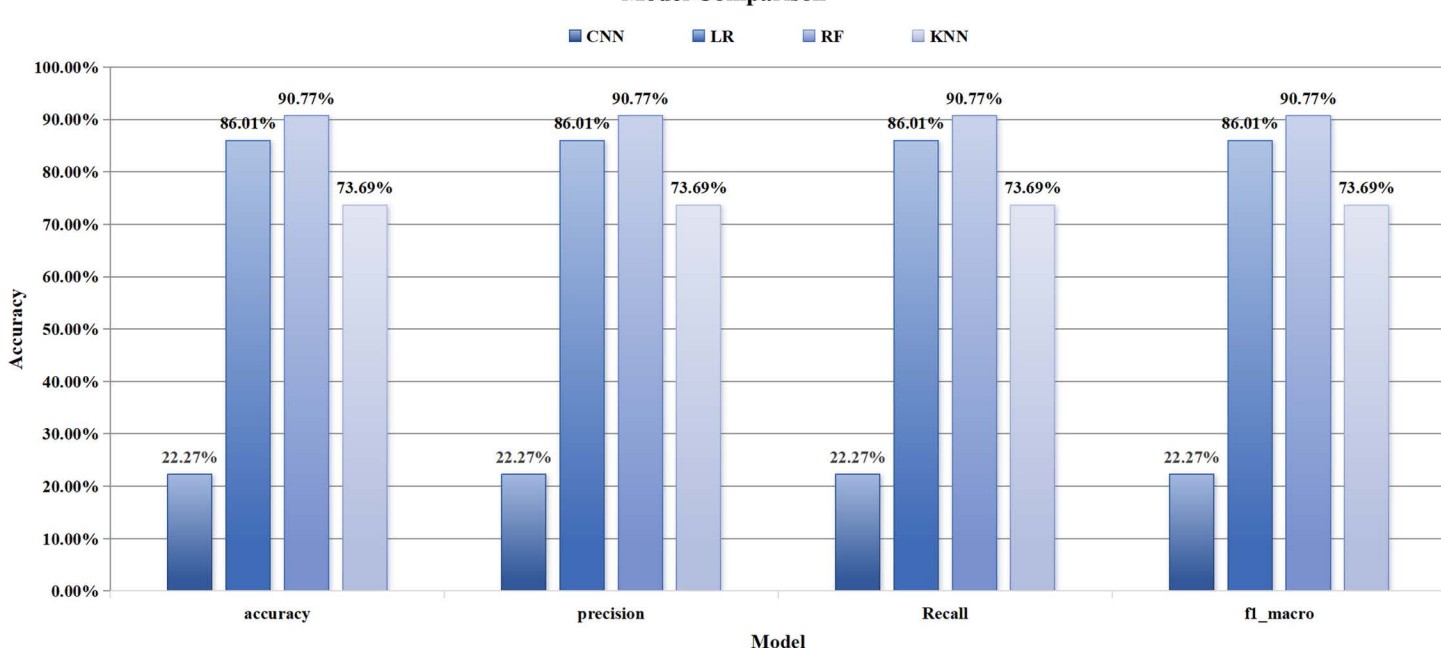

**Fig 6. Comparison of Models.**

entry, including the selection of budget items, takes approximately 120 seconds. The Finance Department's research indicates that under normal circumstances, there are about 50 reimbursements per day. Among these, approximately 8 are returned for modification due to incorrect descriptions or mismatched budget items. This problem is identified after approval by the Finance Department, following the initiator's filling and the department head's approval. Assuming each of these three steps takes 120 seconds, the fastest approval of the identified issue takes 360 seconds. Returning for modification, refilling, and re-approval by the Finance Department results in a total reimbursement filling and approval time of 720 seconds. With 8 reimbursement information approvals per day, the departmental leadership approval time is twice the normal approval time of 120 seconds. The Treasury, with one person responsible for approval, will take 960 seconds (32 minutes) to complete the daily reimbursement information approval at the fastest pace. Assuming an 8-hour workday and the shortest approval process, completing the reimbursement information approval requires 1/16 of the workday. With 22 normal working days per month, this work requires 1.375 days. Assuming a monthly salary of 5,000 yuan per person, the average hourly wage on normal working days is 28.4 yuan, and the normal monthly salary to complete the 3-step approval is 468.6 yuan. Therefore, completing the re-reporting and re-approval requires an additional monthly salary of 312.4 yuan per person, totaling about 937.2 yuan, which is twice the original amount. Using this model and calculating based on an error prediction rate of 9.23%, about 6 out of 50 reimbursement information entries require modification and re-approval. Assuming the shortest 3-step approval process, a year-on-year work efficiency improvement of 25% results in monthly payroll savings of approximately 234.3 yuan. In the intelligent automatic classification of financial reimbursement, the Financial Services Department reports that each telephone consultation on reimbursement takes about 60 seconds. Under normal circumstances, with about 50 reimbursements per day, telephone consultations take one hour per person per day. Calculating based on an 8-hour workday, one person works 7 hours per day in addition to answering the phone. Assuming a monthly salary of 5,000 yuan per person, with 22 normal working days per month and an average hourly salary of about 28.4 yuan on a normal working day, the monthly salary required for telephone counseling is approximately

625 yuan per person. Adopting the present model and calculating based on an incorrect prediction rate of 9.23%, the monthly salary required for telephone counseling is about 57.69 yuan per person. This saves about 567.31 yuan in wage expenditure per person per month and reduces telephone consultations by approximately 10.83 times. Moreover, reducing telephone communication and avoiding the impact of artificial negative emotions contributes to improving the quality of work and life and the physical and mental health of faculty members.

## 4. Discussion

### 4.1. WetNet-RF outperforms common machine learning models in terms of accuracy

This paper introduces the WeNet-RF model for classifying financial reimbursement budget items. By utilizing WeNet, spoken language is effectively converted into text with an impressive accuracy rate of 90.01%, significantly improving data entry efficiency. The original 80-field financial reimbursement dataset was preprocessed, retaining 38 fields for training and validation to improve data quality. For model construction, the newly constructed text dataset containing two fields, summary and budget item, is used to train and validate the model using RF and compare it with three models: CNN, LR, and K-Nearest Neighbors (KNN). Experimental results demonstrate that the RF model performs well in terms of accuracy, precision, recall, and F1 score [33], all these indicators are 90.77%. Fig 7 illustrates the structure of the RF model.

The RF model achieves a precision, recall, and F1 score of 90.77% across 50 real-world data tests, validating its efficacy in genuine business applications. This research provides compelling evidence for the intelligent automation of financial reimbursement processes, which has the potential to deliver substantial economic benefits and efficiency enhancements for enterprises.

### 4.2. Superior performance in automatic classification of radio frequency claims information

The experimental results show that the different models exhibit significant differences in accuracy, precision, recall, and F1 score. The CNN model demonstrates relatively poor performance in processing financial text data due to the limited dataset and inability to adequately capture contextual information, resulting in accuracy, precision, recall, and F1 scores of 22.23%. The LR model, despite its simplicity and ease of implementation, exhibits limited performance in processing high-dimensional and complex financial text data [37], with accuracy, precision, recall, and F1 scores of 86.25%. The performance of the KNN model is influenced by data distribution and similarity metrics, leading to significant fluctuations in its performance, with accuracy, precision, recall, and F1 scores of 56.86%. In contrast, the RF model excels in all aspects. By integrating multiple decision trees, the RF model reduces the risk of overfitting and effectively handles high-dimensional data. In our experiments, the RF model achieves accuracy, precision, recall, and F1 scores of 90.07%, with precision, recall, and F1 scores reaching 90.77% in real data tests, significantly outperforming the other models. Furthermore, the RF model demonstrates robust performance and stability, enabling it to handle financial text classification tasks across various scenarios. Beyond its high accuracy, the RF model also exhibits strong performance in practical applications. Moreover, the RF model's interpretability facilitates understanding and optimization. Future research will focus on further optimizing the RF model to enhance its performance in financial intelligent reimbursement budget item classification.

### 4.3. Evaluation of the effectiveness of real data applications

To fully validate the RF model's performance in authentic financial reimbursement scenarios, we applied it to 50 real-world reimbursement test cases. The primary metric of focus was the model's accuracy rate, which intuitively reflects the proportion of samples predicted as positive by the model that are indeed true positives. In practical applications, a high accuracy rate indicates the model's ability to precisely identify summary texts of financial claims belonging to a specific category, which is essential for avoiding misclassification and potential misuse. In our experiments, the RF model demonstrated a high precision rate of 90.01%, effectively filtering out key information from financial texts and providing robust support for subsequent decision-making processes.

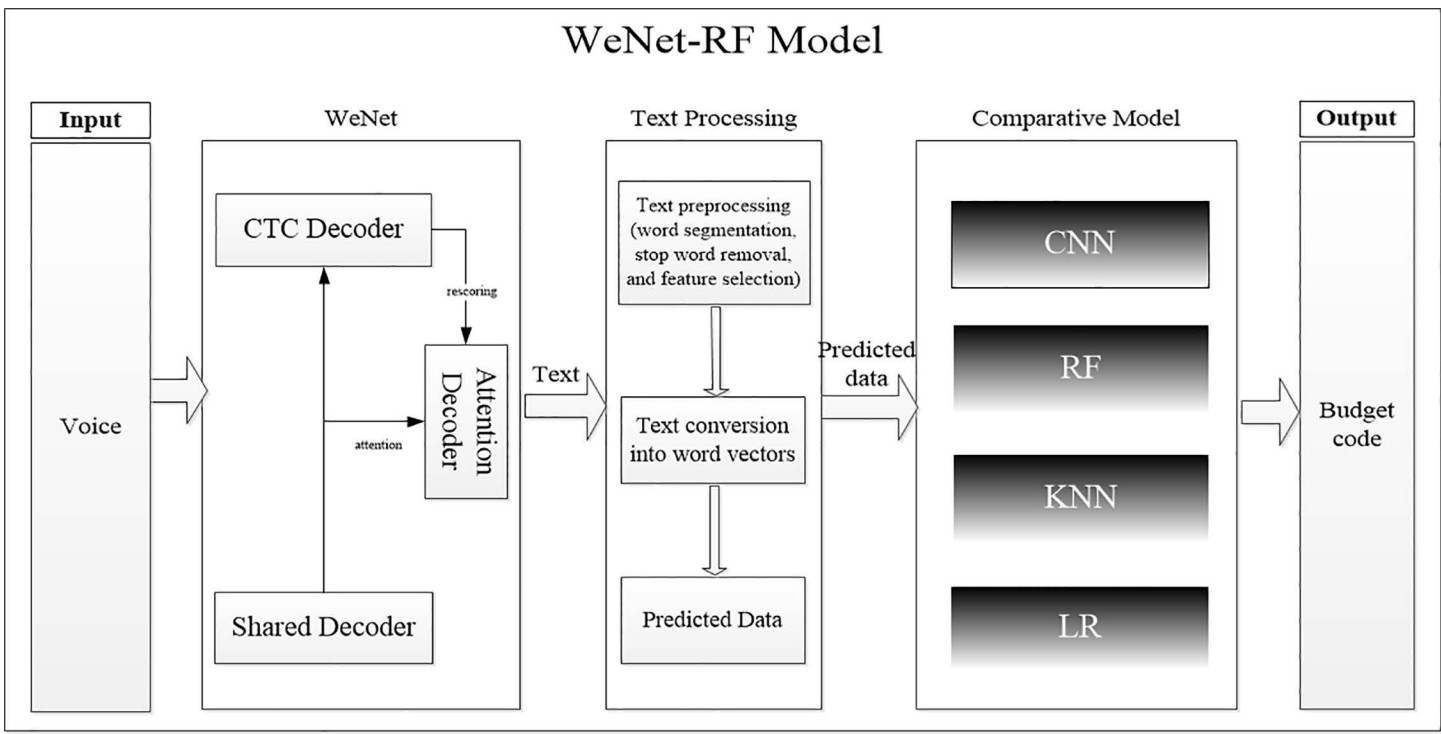

**Fig 7. WeNet-RF model structure.**

Moreover, we conducted a comprehensive analysis of the model's recall [38]. Recall measures the proportion of positive samples that are correctly predicted as positive by the model. In the context of financial reimbursement, a high recall rate indicates the model's ability to maximize the coverage of all relevant financial reimbursement text information and minimize errors caused by omissions [39]. The experimental results demonstrate that the RF model achieves a high recall rate of 90.01%, effectively capturing key information in financial texts and providing comprehensive data support for financial processing. Additionally, we combined precision and recall metrics to calculate the model's F1 score. The F1 score, serving as a harmonized average of precision and recall, offers a more balanced and comprehensive evaluation perspective [40]. The experimental outcomes further validate the RF model's effectiveness and practicality in financial reimbursement scenarios, as evidenced by its excellent F1 score of 90.01%.

In addition to the three primary evaluation metrics discussed above, the RF model demonstrated variability when processing financial text data related to the reimbursement of multiple budget items [41]. This variability may be attributed to the characteristics and distribution of different text categories. To further enhance the model's performance, the data, parameters, and structure of the model were optimized for text data involving reimbursements across multiple budget items. The superiority of the RF model was further validated through its application to real-world scenario data, confirming its effectiveness in this domain.

## 5. Shortcomings of the model

In order to facilitate the rapid entry of financial reimbursement information and ensure the accuracy of budget item classification, several key research directions will be explored. Firstly, the application of WeNet speech recognition for speech-to-text conversion will be studied to enable fast data entry. By optimizing the acoustic model, language model, and decoding algorithms, the aim is to enhance the applicability of the WeNet framework in financial smart bill filling scenarios. Additionally,

the combination of voice and text entry methods will be investigated to provide users with more flexible and convenient input options. Regarding data preprocessing, the focus will be on developing more effective techniques for extracting and cleaning key information from financial text data. Considering the unique characteristics of financial text, smarter data cleaning and feature extraction methods will be researched to minimize the impact of noisy data and improve the training efficiency and performance of the model [42]. Furthermore, efforts will be made to design a more rational data preprocessing workflow that aligns with practical application scenarios and caters to the needs of diverse users. Model optimization is another crucial research area. The objective is to enhance the precision rate, recall rate, and F1 score of the model to further improve its accuracy. This will involve fine-tuning model parameters, optimizing the model architecture, and incorporating novel algorithms and techniques to boost the model's performance in processing complex financial text data [43]. For instance, the integration of deep learning techniques with traditional machine learning algorithms, or the adoption of advanced approaches such as transfer learning and reinforcement learning, may be explored to enhance the model's generalization ability.

## 6. Conclusions

This study presents the WeNet-RF model for classifying financial reimbursement budget items. The experimental results demonstrate that: (1) WeNet effectively facilitates the automatic recognition of financial reimbursement voice content, achieving a recognition accuracy of 90.01%; (2) Data entry speed is significantly improved through WeNet speech recognition, with the three-step approval process expedited by approximately 25%; (3) The proposed RF model supports the automatic classification of reimbursement information, attaining accuracy, precision, recall, and F1 scores all exceeding 90%. This model not only enhances data entry efficiency but also reduces classification errors. Furthermore, it contributes to the intelligentization of the financial sector, promoting the modernization of financial management, with potential applications in other related fields.

## 7. Future work

Future experiments will concentrate on assessing the model's performance using key metrics such as precision, recall, and F1 score, supported by extensive experimental data. A comparative analysis of the model's performance under various optimization strategies will be conducted, alongside an exploration of its applicability across diverse datasets and scenarios. Additionally, rigorous testing and validation in real-world business contexts will be undertaken to ensure the model's effectiveness and stability in practical applications. To enhance model interpretability and traceability, we will conduct research aimed at increasing the transparency and comprehensibility of the models. This will enable users to better understand the decision-making processes involved and facilitate effective monitoring and adjustments. By developing intuitive explanatory metrics and visualization tools, users will gain clearer insights into the model's operations, fostering greater trust and acceptance [44]. Cross-platform and cross-device compatibility will also be a critical focus. Given the increasing diversity of smart devices, it is essential to ensure that models function seamlessly across various platforms. By optimizing deployment methods and interface designs, we aim to achieve robust cross-platform functionality and facilitate collaboration across devices, thus providing users with more convenient and efficient services.

Furthermore, user privacy and security are of paramount importance. In the context of financial smart filling, protecting users' personal information and financial data is crucial. We will conduct in-depth research to develop measures that safeguard user privacy and data security, effectively preventing data breaches and misuse [45]. The implementation of encryption algorithms, security protocols, and privacy protection technologies will help ensure the integrity and confidentiality of user data, thereby providing a reliable service guarantee [46].

In conclusion, this comprehensive research addresses the multifaceted challenges associated with WeNet speech recognition, speech-to-text conversion, text entry, data preprocessing, model optimization, interpretability, traceability, cross-platform compatibility, and user privacy. The overarching goal is to develop more efficient, accurate, and secure automated solutions for the financial reimbursement process.

## Acknowledgements

Not applicable.

## Author contributions

**Conceptualization:** Peichun Suo.

**Data curation:** Wen Suo, Fubing Wang.

**Formal analysis:** Xiuyan Wang.

**Funding acquisition:** Tingting Zeng.

**Methodology:** Peichun Suo.

**Project administration:** Peichun Suo.

**Resources:** Yujing Zhang.

**Software:** Peichun Suo.

**Supervision:** Jinfen Duan.

**Validation:** Meicai Zhu.

**Visualization:** Wen Suo.

**Writing – original draft:** Peichun Suo.

**Writing – review & editing:** Xiuyan Wang, Weili Kou.

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
