## [Decision Letter · Decision Letter 0]

9 Feb 2025

PONE-D-24-45687Automatic Classification Model for Financial Reimbursement Budget Lines Based on WeNet-Random ForestPLOS ONE

Dear Dr. Suo,

Thank you for submitting your manuscript to PLOS ONE. After careful consideration, we feel that it has merit but does not fully meet PLOS ONE’s publication criteria as it currently stands. Therefore, we invite you to submit a revised version of the manuscript that addresses the points raised during the review process.

We look forward to receiving your revised manuscript.

Kind regards,

Arnold Adimabua Ojugo, PhD

Academic Editor

PLOS ONE

Journal Requirements:

3. In the online submission form, you indicated that “Data cannot be shared publicly due to the high security requirements of financial information. For researchers who meet the criteria for accessing confidential data, they can obtain the data by contacting the author via email.”

4. We are unable to open your Supporting Information file “Supporting Information.rar”. Please kindly revise as necessary and re-upload.

**Additional Editor Comments:**

Authors to revised as suggested

Reviewers' comments:

Reviewer's Responses to Questions

**Comments to the Author**

1. Is the manuscript technically sound, and do the data support the conclusions?

Reviewer #1: Yes

Reviewer #2: Partly

2. Has the statistical analysis been performed appropriately and rigorously? 

Reviewer #1: Yes

Reviewer #2: I Don't Know

3. Have the authors made all data underlying the findings in their manuscript fully available?

Reviewer #1: Yes

Reviewer #2: Yes

4. Is the manuscript presented in an intelligible fashion and written in standard English?

Reviewer #1: Yes

Reviewer #2: Yes

5. Review Comments to the Author

Reviewer #1: 1- The researcher did not explain in the abstract the research gap as well as the steps of the proposed system in a simple manner.

2- Clarifying the goal and the extent to which it was achieved from each point contributed by the research. In addition to adding a paragraph at the end of the introduction that clarifies the structure of the research in all its sections.

3- Are there techniques for converting or extracting texts before passing them to the proposed system? Also, has the data set been included in a specific location to use a link and include it in the references?

4- Comparing the results with other previous works according to the approved standards

5- Include references published in 2024, with no less than 4 references.

Reviewer #2: The manuscript describes a technical solution real world problem. The research was described with supporting data that yields the conclusions made. The experimental procedure was also described much like a literature. However, the technical definitions of the model controls and parameter tuning were not shown. Also, the model training process and the number of epochs was not described. The simulation experiments and the number of experiments conducted were not also described. What was show is the mean values of the experiment.

I cannot ascertain if the right statistical procedures were followed and rigorously carried out. Though, literary description were presented; there was not technical details to support and validate those descriptions in terms of the model equations and simulation experiments performed and how values were captured/generated from such experiments.

The sample data used for the research were described, stating how they were acquired, the dataset size as also stated. The number of data points used by the model for it plots were also specified. The description of the model evaluation parameters was adequately described with appropriate comparison as stated in the object of the manuscript. However, authors required access to research data through am email request due to the highly sensitive nature of financial data.

The manuscript was well presented in a logical and intelligent manner, clearly written in standard English Language which was clearly understood. However. the reporting is required to be done in past tense since the research has been completed. Some acronyms that were uses at first instance (i.e. MODIS, ASR, etc.) were not were in full.

6. PLOS authors have the option to publish the peer review history of their article (what does this mean? ). If published, this will include your full peer review and any attached files.

**Do you want your identity to be public for this peer review?** For information about this choice, including consent withdrawal, please see our Privacy Policy .

Reviewer #1: No

Reviewer #2: **Yes: ** Onate Egerton Taylor

---

## [Author Response · Author response to Decision Letter 1]

24 Feb 2025

Manuscript Number PONE-D-24-45687

Automatic Classification Model for Financial Reimbursement Budget Items Based on WeNet-Random Forest

Dear Editor of PLoS One,

Thank you very much for taking the time to review this manuscript. I truly appreciate all your comments and suggestions! After carefully considering the revision suggestions, we have revised the title of the paper again to: "WeNet-RF: An Automatic Classification Model for Financial Reimbursement Budget Items".Below are my itemized responses, and the resubmitted files contain the journal requirements as well as my revisions to the manuscript. We will respond to the comments of the two reviewers point by point separately. The requirements of the journal are as follows.

Question 1: Please ensure that your manuscript meets PLOS ONE's style requirements, including those for file naming. The PLOS ONE style templates can be found at https://journals.plos.org/plosone/s/file?id=wjVg/PLOSOne_formatting_sample_main_body.pdf and https://journals.plos.org/plosone/s/file?id=ba62/PLOSOne_formatting_sample_title_authors_affiliations.pdf

Response: I have carefully checked and revised the manuscript in accordance with the formatting requirements of PLOS ONE.

Question 2: We note that the grant information you provided in the ‘Funding Information’ and ‘Financial Disclosure’ sections do not match. When you resubmit, please ensure that you provide the correct grant numbers for the awards you received for your study in the ‘Funding Information’ section.

Response: We apologize for the discrepancy between the ‘Funding Information’ and ‘Financial Disclosure’ sections. We have carefully reviewed the grant details related to our study and have made the necessary corrections to ensure consistency. The correct grant numbers for the awards we received are now accurately reflected in the ‘Funding Information’ section. We have also double-checked this information against our records to avoid any further inconsistencies.

Question 3: In the online submission form, you indicated that “Data cannot be shared publicly due to the high security requirements of financial information. For researchers who meet the criteria for accessing confidential data, they can obtain the data by contacting the author via email.”

Response: We understand the importance of transparency and accessibility in research. However, due to the highly sensitive nature of the financial information included in our study, the data cannot be shared publicly. This is to ensure compliance with strict security and confidentiality protocols that are essential for protecting the integrity of the financial data involved.For researchers who meet the criteria for accessing confidential data, we are willing to facilitate data sharing on a case-by-case basis. Interested researchers can directly contact the school's financial teacher via email according to the email address provided in the thesis to apply for access to the data. Then, we will evaluate each application based on the established criteria and security requirements.

Question 4: All PLOS journals now require all data underlying the findings described in their manuscript to be freely available to other researchers, either 1. In a public repository, 2. Within the manuscript itself, or 3. Uploaded as supplementary information.

This policy applies to all data except where public deposition would breach compliance with the protocol approved by your research ethics board. If your data cannot be made publicly available for ethical or legal reasons (e.g., public availability would compromise patient privacy), please explain your reasons on resubmission and your exemption request will be escalated for approval.We are unable to open your Supporting Information file “Supporting Information.rar”. Please kindly revise as necessary and re-upload.

Response: Regarding the file previously titled "Supporting Information.rar", I have renamed and re-uploaded it as "Supplementary_Main_program_code.pdf".

Question 5: Please review your reference list to ensure that it is complete and correct. If you have cited papers that have been retracted, please include the rationale for doing so in the manuscript text, or remove these references and replace them with relevant current references. Any changes to the reference list should be mentioned in the rebuttal letter that accompanies your revised manuscript. If you need to cite a retracted article, indicate the article’s retracted status in the References list and also include a citation and full reference for the retraction notice.

Response: I have thoroughly reviewed all the references cited in our manuscript to ensure their accuracy and completeness. I can confirm that there are no citations of retracted papers in our reference list. We have also double-checked the references to ensure that they are up-to-date and relevant to our study. No changes were necessary, as all references are accurate and appropriate for the context of our research.

Dear Reviewer,

Thank you very much for your valuable comments and suggestions. We have carefully considered each point and made the necessary revisions to improve the manuscript. At the same time, we have changed the title of the paper to: "WeNet-RF: An Automatic Classification Model for Financial Reimbursement Budget Items". We hope that the revised manuscript addresses all of your concerns. Below are our detailed responses to each of your comments:

Question 1: The researcher did not explain in the abstract the research gap as well as the steps of the proposed system in a simple manner.

Response 1: We have thoroughly revised the abstract to clearly articulate the research gap and the steps involved in our proposed system. The revised abstract now provides a concise yet comprehensive overview of the study's motivation, methodology, and contributions.

Question 2: Clarifying the goal and the extent to which it was achieved from each point contributed by the research. In addition to adding a paragraph at the end of the introduction that clarifies the structure of the research in all its sections.

Response 2: We appreciate your guidance on this matter. We have added a dedicated paragraph at the end of the introduction to clearly outline the research goals and the extent to which they were achieved. Additionally, we have provided a detailed overview of the manuscript structure to guide readers through each section. This change ensures that the research objectives and the overall organization of the study are transparent to the readers.

Question 3: Are there techniques for converting or extracting texts before passing them to the proposed system? Also, has the data set been included in a specific location to use a link and include it in the references?

Response 3: We have added a detailed description of the text preprocessing techniques in Section 2.1.3 (Data Preprocessing), specifying the methods used for text conversion and extraction. Regarding the data set, we have clarified the data acquisition process in Section 2.1.1 (Data Sources Description). Due to the sensitive nature of financial data, we have provided instructions for accessing the dataset via email request, as indicated in the "Data Availability Statement" section.

Question 4: Comparing the results with other previous works according to the approved standards.

Response 4: We understand the importance of comparing our results with existing literature. However, after conducting an extensive literature review, we found that there are no published studies that apply the Random Forest algorithm specifically to the classification of financial budget items in private schools. As such, a direct comparison with similar models is not feasible. Nevertheless, we have ensured that our methodology and results are presented in a manner that aligns with the best practices in the field.

Question 5: Include references published in 2024, with no less than 4 references.

Response 5: We have conducted additional research and incorporated several recent publications from 2024 to enhance the relevance and timeliness of our manuscript. Specifically, we have cited the following studies:

Moritaka and Komuro (2024) proposed a two-layer Random Forest model based on opcode sequences to enhance ransomware detection.

Sun et al. (2024) improved the Random Forest algorithm by optimizing classification accuracy and decision tree correlation.

Liu, Li, and Yang (2024) applied a multi-algorithm hybrid feature extraction model to underwater acoustic signals.

Bigot, Dabo, and Male (2024) conducted a high-dimensional analysis of ridge regression for non-independent and non-identically distributed data.

These references provide valuable context and support for our research.

Dear Reviewer,

Thank you very much for your detailed and constructive feedback. We have carefully reviewed your comments and made the necessary revisions to address each concern. At the same time, we have changed the title of the paper to: "WeNet-RF: An Automatic Classification Model for Financial Reimbursement Budget Items". Below are our detailed responses:

Question 1: The manuscript describes a technical solution to a real-world problem. However, the technical definitions of model controls and parameter tuning were not shown. Additionally, the model training process and the number of epochs were not described. The simulation experiments and the number of experiments conducted were also not described. Only the mean values of the experiments were shown.

Response 1: We have added detailed descriptions of the hyperparameter tuning process in Section 2.6 (Random Forest Model). Specifically, we have included information on the tree depth and other critical parameters used in the Random Forest model (Lines 292-294). In Section 2.7 (Cross-Validation Assessment), we have clarified the number of simulation experiments conducted and the statistical methods used to analyze the results (Lines 317-318). These additions provide a comprehensive understanding of the model's training and evaluation processes.

Question 2: I cannot ascertain if the right statistical procedures were followed and rigorously carried out. Though literary descriptions were presented, there were no technical details to support and validate those descriptions in terms of model equations and simulation experiments.

Response 2: We understand the importance of rigorous statistical validation. To address this, we have provided detailed descriptions of the statistical methods used in the study, including the equations and simulation experiments. This information is now included in the relevant sections of the manuscript, ensuring that readers can fully understand and reproduce our results.

Question 3: The sample data used for the research were described, stating how they were acquired and the dataset size. However, the authors require access to research data through an email request due to the highly sensitive nature of financial data.

Response 3: We acknowledge the journal's policy on data availability. While we understand the importance of making data publicly available, the financial data used in this study are highly sensitive and cannot be shared publicly. We have added a clear statement in the "Data Availability Statement" section, indicating that the dataset can be obtained by contacting the teachers of the school's financial department via email (Name: Zeng Tingting; Contact number: +86 871-68305391; Email address: 1908584074@qq.com). We are committed to facilitating access to these data for researchers who meet the necessary criteria.

Question 4: The manuscript was well presented in a logical and intelligent manner, clearly written in standard English. However, the reporting is required to be done in the past tense since the research has been completed. Some acronyms that were used at first instance (i.e., MODIS, ASR, etc.) were not expanded in full.

Response 4: We have carefully reviewed the manuscript and revised the language to ensure that all descriptions are in the past tense, reflecting the completed nature of the research. Additionally, we have expanded all acronyms at their first appearance to avoid confusion for readers. These changes enhance the clarity and consistency of the manuscript.

We hope that these revisions address all of your concerns. We appreciate your valuable feedback and look forward to your positive response.

Thank you once again for your time and consideration.

Best regards,

Peichun Suo

---

## [Decision Letter · Decision Letter 1]

3 Mar 2025

WeNet-RF: An Automatic Classification Model for Financial Reimbursement Budget Items

PONE-D-24-45687R1

Dear Dr. Suo,

We’re pleased to inform you that your manuscript has been judged scientifically suitable for publication and will be formally accepted for publication once it meets all outstanding technical requirements.

Kind regards,

Arnold Adimabua Ojugo, PhD

Academic Editor

PLOS ONE

Additional Editor Comments (optional):

Reviewers' comments:

Reviewer's Responses to Questions

**Comments to the Author**

1. If the authors have adequately addressed your comments raised in a previous round of review and you feel that this manuscript is now acceptable for publication, you may indicate that here to bypass the “Comments to the Author” section, enter your conflict of interest statement in the “Confidential to Editor” section, and submit your "Accept" recommendation.

Reviewer #1: All comments have been addressed

Reviewer #2: (No Response)

2. Is the manuscript technically sound, and do the data support the conclusions?

Reviewer #1: (No Response)

Reviewer #2: Yes

3. Has the statistical analysis been performed appropriately and rigorously? 

Reviewer #1: (No Response)

Reviewer #2: N/A

4. Have the authors made all data underlying the findings in their manuscript fully available?

Reviewer #1: (No Response)

Reviewer #2: Yes

5. Is the manuscript presented in an intelligible fashion and written in standard English?

Reviewer #1: (No Response)

Reviewer #2: Yes

6. Review Comments to the Author

Reviewer #1: (No Response)

Reviewer #2: In the Abstract, that model value for the performance parameters is not define (i.e. exceeding 90%), for a scientific research with result that is represented quantitatively, it is expected that the result value is definite or specified as a range. Please, adequately address.

The value of the Random Forest model parameters in Table 3 is specified as 90.77 as opposed 92.77 as specified in lines: 359, 425, and 443. Please clarify.

The descriptions of model equations and various runs of the simulation experiments are yet not available in the revised manuscript.

7. PLOS authors have the option to publish the peer review history of their article (what does this mean? ). If published, this will include your full peer review and any attached files.

**Do you want your identity to be public for this peer review?** For information about this choice, including consent withdrawal, please see our Privacy Policy .

Reviewer #1: No

Reviewer #2: **Yes: ** Onate Egerton Taylor

---

## [Editor Report · Acceptance letter]

PONE-D-24-45687R1

PLOS ONE

Dear Dr. Suo,

I'm pleased to inform you that your manuscript has been deemed suitable for publication in PLOS ONE. Congratulations! Your manuscript is now being handed over to our production team.

Kind regards,

on behalf of

Prof. Arnold Adimabua Ojugo

Academic Editor

PLOS ONE